# Channel-Hopping Sequence and Searching Algorithm for Rendezvous of Spectrum Sensing

**DOI:** 10.3390/s25010062

**Published:** 2024-12-25

**Authors:** Young-June Choi, Young-Sik Kim, Ji-Woong Jang

**Affiliations:** 1Department of Software and Computer Engineering, Ajou University, Suwon 16499, Republic of Korea; choiyj@ajou.ac.kr; 2Department of Electrical Engineering and Computer Science, Daegu Gyeongbuk Institute of Science and Technology, Daegu 42988, Republic of Korea; ysk@dgist.ac.kr; 3Department of Computer Engineering, Ulsan College, Ulsan 44022, Republic of Korea

**Keywords:** spectrum sensing, cognitive radio networks (CRNs), blind rendezvous, asymmetric channel model, sequence, fast randezvous modes

## Abstract

In this paper, we propose a method for applying the *p*-ary m-sequence as a channel-searching pattern for rendezvous in the asymmetric channel model of cognitive radio. We mathematically analyzed and calculated the ETTR when the m-sequence is applied to the conventional scheme, and our simulation results demonstrated that the ETTR performance is significantly better than that of the JS algorithm. Furthermore, we introduced a new channel-searching scheme that maximizes the benefits of the m-sequence and proposed a method to adapt the generation of the m-sequence for use in the newly proposed scheme. We also derived the ETTR mathematically for the new scheme with the m-sequence and showed through simulations that the performance of the new scheme with the m-sequence is superior to that of the conventional scheme with the m-sequence. Notably, when there is only one common channel, the new scheme with the m-sequence achieved approximately four times the improvement in the ETTR compared to the conventional scheme.

## 1. Introduction

Cognitive radio technology has emerged as a promising solution to the problem of inefficient spectrum utilization. One of the fundamental challenges in cognitive radio networks is the rendezvous problem, which involves enabling secondary users to establish a common communication channel without prior coordination. The efficiency of channel rendezvous is critical, particularly in environments with asymmetric channel availability, where the distribution and accessibility of channels vary between devices [1,2,3].

Previous research has explored various strategies to address the rendezvous problem in asymmetric environments. Algorithms like ISAC construct channel-hopping sequences using only available channels to reduce the expected time to rendezvous (ETTR) [4]. Additionally, *p*-ary m-sequence-based channel hopping has been analyzed for its performance in a symmetric channel model [5]. Jump-stay-based algorithms, such as the one proposed by Lin and colleagues, have also been widely studied for their ability to guarantee rendezvous by alternating between “jump” and “stay” patterns, minimizing time-to-rendezvous even in asynchronous scenarios [6]. These methods ensure robust performance, particularly in environments where the channel availability differs significantly between devices.

Recent studies have also emphasized the importance of robust channel-hopping patterns in practical applications. For example, the integration of cognitive radio principles with wireless sensor networks (WSNs) has demonstrated the significance of dynamic spectrum access in ensuring reliable communication in distributed environments [2]. Advances in resource allocation strategies and sequence optimization have further contributed to enhancing efficiency and mitigating interference in CR networks [3]. Moreover, the application of CR principles in IoT scenarios has highlighted scalable approaches to address the challenges of densely connected networks [4].

Further advancements in rendezvous strategies have focused on systematically constructing robust channel-hopping sequences that ensure rendezvous in both symmetric and asymmetric channel models [7]. For instance, composite channel-hopping algorithms have been developed to address the challenges posed by heterogeneous cognitive radio networks, particularly in blind rendezvous scenarios [8,9]. Additionally, novel asynchronous channel-hopping schemes have demonstrated their ability to achieve fair and fast rendezvous without relying on time synchronization, even under dynamic channel availability [10]. Finally, quinary coding and matrix-based approaches have been introduced to guarantee rendezvous under diverse and asynchronous channel conditions, providing scalable and reliable performance [11].

Building on these established approaches, we focus on applying *p*-ary m-sequences as a channel-hopping strategy specifically designed for asymmetric channel environments. The *p*-ary m-sequence, known for its desirable autocorrelation properties and even distribution across channels, provides a strong foundation for efficient rendezvous mechanisms. Furthermore, we extend our research to scenarios where the search space is reduced. By strategically limiting the number of channels to be searched and applying *p*-ary m-sequences, our approach aims to further minimize the ETTR, enhancing system performance in cognitive radio networks.

Our study contributes to the field by proposing and analyzing a novel channel-searching scheme tailored to asymmetric channel conditions. We demonstrate that our method not only retains the advantages of traditional *p*-ary m-sequence strategies but also benefits from insights drawn from jump-stay-based algorithms, achieving significant improvements in rendezvous efficiency. This work holds practical implications for designing faster and more reliable cognitive radio systems, providing new insights into efficient channel hopping in complex, asymmetric environments.

The remainder of this paper is organized as follows: In Section 2, we analyze the performance of the *p*-ary m-sequence as a channel-hopping sequence in an asymmetric channel model. Section 3 proposes a new channel-searching scheme for rendezvous and analyzes the performance of the *p*-ary m-sequence for this suggested scheme. Section 4 presents the numerical results of the proposed scheme. Finally, Section 5 concludes the paper and suggests directions for future research.

## 2. Applying *p*-Ary m-Sequence to Asymmetric Channel Model

In this section, we propose using the *p*-ary m-sequence as a channel-hopping sequence for achieving rendezvous in the asymmetric channel model. We also calculate the Expected Time to Rendezvous (ETTR) for the conventional scheme, applying the *p*-ary m-sequence as a channel-searching pattern.

In the asymmetric model, two users may have different sets of available channels. The primary approach for achieving rendezvous in this model involves extending each user’s available channel set to cover all possible channels and then applying a channel-hopping pattern over the entire set. However, because the available sets are expanded to include all channels, the two users may attempt to rendezvous on a channel that is unavailable to one or both of them, thereby preventing a successful rendezvous. To address this issue, Lin, Liu, Chu, and Leung proposed a method that ensures rendezvous over all possible channels using a carefully designed channel-hopping pattern. Unlike in the symmetric model, where rendezvous can occur if any common channel is selected, in the asymmetric model, rendezvous only succeeds when both users are aligned on a channel that is available to both.

Let *p* be a prime and *n* be a positive integer. Let $F_{p^{n}}$ be the finite field with $p^{n}$ elements. Then, the trace function ${\mathrm{tr}}_{1}^{n}\left(x\right)$ from $F_{p^{n}}$ to $F_{p}$ is defined as
$$\begin{array}{c} {\mathrm{tr}}_{1}^{n}\left(x\right)=\sum_{t=1}^{n}x^{p^{i}} \end{array}$$
where $x\in F_{p^{n}}$. Then, the trace function satisfies the following properties

(1)${\mathrm{tr}}_{1}^{n}\left(ax+by\right)=a{\mathrm{tr}}_{1}^{n}\left(x\right)+b{\mathrm{tr}}_{1}^{n}\left(y\right)$, $a,b\in F_{p}$ and $x,y\in F_{p^{n}}$(2)${\mathrm{tr}}_{1}^{n}\left(x^{p}\right)={\mathrm{tr}}_{1}^{n}\left(x\right)$, for all $x\in F_{p^{n}}$.

Using the definition of trace function, we can define the *p*-ary m-sequence m(t) of the period $p^{n}-1$ as follows:$$m\left(t\right)={\mathrm{tr}}_{1}^{n}\left(\alpha^{t}\right)$$
where α is a primitive element of $F_{p^{n}}$. It is well known that the *p*-ary m-sequence m(t) has a balance property such that
$$\begin{array}{c} m\left(t\right)=\left\{\begin{array}{cc} k, & p^{n-1}\;\mathrm{times}\;\mathrm{for}\;1\leq k\leq p-1,\\ 0, & p^{n-1}-1\;\mathrm{times}. \end{array}\right.\; \end{array}$$

Let *p* be a prime and n≥1 be a positive integer. We use the notation that $F_{p^{n}}$ is the finite field with $p^{n}$ elements and $F_{p^{n}}^{*}=F_{p^{n}}/\left\{0\right\}$. In the remainder of the paper, we assume that p>2. Following is a generalization of the balance property and the two-level ideal autocorrelation function property of *p*-ary m-sequences. Note that the period of a sequence is always the minimum period of the sequence in this paper

**Definition 1** (Balanced and Difference-balanced [12]). *A q-ary sequence s(t) of period $q^{n}-1$ is said to be balanced if zero appears $q^{n-1}-1$ times and any non-zero element of $F_{q}$ appears $q^{n-1}$ times in one period. It is said to be difference-balanced [13,14] if, for any non-zero τ mod $q^{n}-1$, in the differences s(t+τ)−s(t) as t runs from 0 to $q^{n}-2$, the value zero occurs $q^{n-1}-1$ times and each of the non-zero values of $F_{q}$ occurs $q^{n-1}$ times.*

**Theorem 1.** 
*For positive integers M and p such that p is the smallest prime bigger than M. Let m(t) be a p-ary m-sequence with period $p^{2}-1$. Using m(t) as a channel-hopping sequence for the cognitive radio with M asymmetric channels, the rendezvous occurs within $p^{2}-1$ with probability $\left(p^{2}-p+1\right)/\left(p^{2}-1\right)$.*


**Proof.** From the balance property of the *p*-ary m-sequence, the number of zeroes of m(t) within a period is p−1. Using the property of the trace function, we have
$$\begin{array}{ccc} {\mathrm{tr}}_{1}^{2}\left(\alpha^{t}\right)-{\mathrm{tr}}_{1}^{2}\left(\alpha^{t+\tau }\right) & = & {\mathrm{tr}}_{1}^{2}\left(\alpha^{t}\left(1-\alpha^{\tau }\right)\right)\\ & = & {\mathrm{tr}}_{1}^{2}\left(\alpha^{t}\alpha^{\delta }\right)\\ & = & {\mathrm{tr}}_{1}^{2}\left(\alpha^{t+\delta }\right) \end{array}$$
where $\alpha^{\delta }=1-\alpha^{\tau }$. Therefore, zero appears p−1 times within a period $p^{2}-1$. Let $T=\left(p^{2}-1\right)/\left(p-1\right)=p+1$, then we have
(1)$${\mathrm{tr}}_{1}^{2}\left(\alpha^{kT+t}\right)=\alpha^{kT}{\mathrm{tr}}_{1}^{2}\left(\alpha^{t}\right).$$
and from the property of the finite field, it is clear that $\alpha^{kT}\in F_{p}^{*}=Z_{p}^{*}$.Let integer $0\leq i<p^{2}-1$ satisfy ${\mathrm{tr}}_{1}^{2}\left(\alpha^{i}\right)=0$. Then, from the above equation, we have
$${\mathrm{tr}}_{1}^{2}\left(\alpha^{kT+i}\right)=0,\;\;for\;0\leq k\leq p$$
where $0\leq kT+i<p^{2}-1$.For different $k_{1}$ and $k_{2}$, $0\leq k_{1},k_{2}<p-1$, let *i* be an integer such that $0\leq k_{1}T+i,k_{2}T+i<p^{2}-1$. From the balance property of m-sequence, m(t) has p−1 zeroes within a period. Therefore, all zeroes of m(t) can be represented as kT+i and this means m(t) has zero within T=p+1 duration, which means m(t) and m(t+τ) has the same number with the same position within the first T=p+1 duration. We can describe the value of the trace function ${\mathrm{tr}}_{1}^{2}\left(\alpha^{t}\right),\;0\leq t<p^{2}-1$ as follows:
(2)$$\begin{array}{cccccc} {\mathrm{tr}}_{1}^{2}\left(\alpha^{0}\right) & {\mathrm{tr}}_{1}^{2}\left(\alpha^{1}\right) & {\mathrm{tr}}_{1}^{2}\left(\alpha^{2}\right) & \cdots & {\mathrm{tr}}_{1}^{2}\left(\alpha^{p-1}\right) & {\mathrm{tr}}_{1}^{2}\left(\alpha^{p}\right)\\ \alpha^{T}{\mathrm{tr}}_{1}^{2}\left(\alpha^{0}\right) & \alpha^{T}{\mathrm{tr}}_{1}^{2}\left(\alpha^{1}\right) & \alpha^{T}{\mathrm{tr}}_{1}^{2}\left(\alpha^{2}\right) & \cdots & \alpha^{T}{\mathrm{tr}}_{1}^{2}\left(\alpha^{p-1}\right) & \alpha^{T}{\mathrm{tr}}_{1}^{2}\left(\alpha^{p}\right)\\ \alpha^{2T}{\mathrm{tr}}_{1}^{2}\left(\alpha^{0}\right) & \alpha^{2T}{\mathrm{tr}}_{1}^{2}\left(\alpha^{1}\right) & \alpha^{2T}{\mathrm{tr}}_{1}^{2}\left(\alpha^{2}\right) & \cdots & \alpha^{2T}{\mathrm{tr}}_{1}^{2}\left(\alpha^{p-1}\right) & \alpha^{2T}{\mathrm{tr}}_{1}^{2}\left(\alpha^{p}\right)\\ \; & \; & & \cdots & \; & \\ \alpha^{(p-3)T}{\mathrm{tr}}_{1}^{2}\left(\alpha^{0}\right) & \alpha^{(p-3)T}{\mathrm{tr}}_{1}^{2}\left(\alpha^{1}\right) & \alpha^{(p-3)T}{\mathrm{tr}}_{1}^{2}\left(\alpha^{2}\right) & \cdots & \alpha^{(p-3)T}{\mathrm{tr}}_{1}^{2}\left(\alpha^{p-1}\right) & \alpha^{(p-3)T}{\mathrm{tr}}_{1}^{2}\left(\alpha^{p}\right)\\ \alpha^{(p-2)T}{\mathrm{tr}}_{1}^{2}\left(\alpha^{0}\right) & \alpha^{(p-2)T}{\mathrm{tr}}_{1}^{2}\left(\alpha^{1}\right) & \alpha^{(p-2)T}{\mathrm{tr}}_{1}^{2}\left(\alpha^{2}\right) & \cdots & \alpha^{(p-2)T}{\mathrm{tr}}_{1}^{2}\left(\alpha^{p-1}\right) & \alpha^{(p-2)T}{\mathrm{tr}}_{1}^{2}\left(\alpha^{p}\right) \end{array}.$$Let $t_{1}$ such that $0\leq t_{1}\leq p$ and $m\left(t_{1}\right)=m\left(t_{1}+\tau \right)$ . From (Equation 2), it is clear that
(3)$$\left\{\alpha^{kT}{\mathrm{tr}}_{1}^{2}\left(\alpha^{t_{1}}\right)|0\leq k<p-1\right\}=\left\{\begin{array}{cc} F_{p}^{*}, & \mathrm{for}\;m(t_{1})\neq 0\\ 0, & \mathrm{for}\;m(t_{1})=0 \end{array}\right..$$Let $0\leq t_{a}<p+1$ be an integer such that $m\left(t_{a}\right)=m\left(t_{\tau }\right)$. In order to calculate the expected TTR, we should consider the following cases for $0\leq \tau <p^{2}-1$ and $m\left(t\right)\in F_{p}$:**Case 1)** τ=0 Once:This case occurs only once for $p^{2}-1$. cases. From the property of the trace function, it is clear that every number in $F_{p}$ is aligned for both users within p+1 time slots. Therefore, rendezvous occurred within p+1 time slots in this case.**Case 2)** τ≠0 and $m(t_{a})=0$ p−2 times:From equation (Equation 3), we can obtain $\alpha^{kT}{\mathrm{tr}}_{1}^{2}\left(\alpha^{t_{a}}\right)=0,\;\mathrm{for}\;0\leq k<p-1$. That means values of all points align with period T=p+1 for two sequence m(t) and m(t+τ) are 0. To make rendezvous for this case, both devices use symbol 0 for their common channel. But this violates the assumption of cognitive radio that it does not have a control channel. Therefore, rendezvous does not occur in this case.**Case 3)** τ≠0 and $m(t_{a})\neq 0$ $p^{2}-p$ times:From the property of the trace function and the assumption of this case, we have $t_{a}$ such that $0\leq t_{a}<p+1$ and the value of $m(t_{a})\neq 0$. From the Equation (Equation 3), it is clear that $\left\{\alpha^{kT}{\mathrm{tr}}_{1}^{2}\left(\alpha^{t_{1}}\right)|0\leq k<p-1\right\}=F_{p}^{*}$, which means two devices align with every channel in the channel model. Therefore, rendezvous occurred within $p^{2}-1$ in this case.From the result of the above cases, rendezvous occurs $p^{2}-p+1$ times out of a total of $p^{2}-1$ cases. In other words, using the method described above, the rendezvous occurs with a probability of $\left(p^{2}-p+1\right)/\left(p^{2}-1\right)$.   □

From the result of the above theorem, we can derive the average time to rendezvous of the proposed scheme if the time delay between the two devices is not a multiple of T=p+1 as follows:

**Theorem 2.** 
*For positive integers M and p such that p is the smallest prime bigger than M. Let m(t) be a p-ary m-sequence with period $p^{2}-1$ and C be the number of common channels between both devices. Using m(t) as the channel-hopping sequence for the cognitive radio with M asymmetric channels, the average TTR of this scheme is $\frac{2\left(p+1\right)+p\left(p-1\right)\left(p^{2}-Mp-M-1\right)}{2M(p^{2}-1)}\approx \frac{p(p-1-M)}{2M}$ except for the time difference of the two devices which is not a multiple of p+1.*


**Proof.** From the result of Theorem 1 and the assumption of this theorem, the ETTR of each case of Theorem 1 can be calculated as follows:**Case 1)** τ=0 Once:In this case, it is clear that channel alignment occurs *M* times within 1 time slots. Therefore, the ETTR of this case is (p+1)/M.**Case 2)** τ≠0 and $m(t_{a})=0$ p−2 times:In this case, randezvous does not occur. Therefore, we should make another trial to rendezvous for this case.**Case 3)** τ≠0 and $m(t_{a})\neq 0$ $p^{2}-p$ times:Let us divide $p^{2}-1$ time slots into p−1 sets with p+1 time slots. In this case, every p−1 set has a different aligned channel in $F_{p}^{*}$ and the average time to randezvous of each time set is (p+1)/2. Since each time set has an equal probability of rendezvous, it is clear that the probability of randezvous for each set is M/(p−1). Therefore, the average time to rendezvous for this case can be calculated as follows:
$$\frac{p^{2}-1}{M}-\frac{p+1}{2}=\frac{p^{2}-Mp-M-1}{2M}.$$From the result of the above cases, the average value of the rendezvous with failure probability $\left(p-2\right)/\left(p^{2}-1\right)$ is
$$\begin{array}{ccc} \frac{1}{p^{2}-1}\times \frac{p+1}{M}+\frac{p}{p+1}\times \frac{p^{2}-Mp+M-1}{2M} & = & \frac{2\left(p+1\right)+p\left(p-1\right)\left(p^{2}-Mp-M-1\right)}{2M(p^{2}-1)}\\ & = & \frac{2\left(p+1\right)+p\left(p-1\right)(p^{2}-1-M\left(p+1\right))}{2M(p^{2}-1)}\\ & = & \frac{2(p+1)+p(p-1)(p+1)(p-1-M)}{2M(p-1)(p+1)}\\ & = & \frac{2+p(p-1)(p-1-M)}{2M(p-1)}\\ & \approx & \frac{p(p-1-M)}{2M} \end{array}$$
except where the time difference between the two devices is not a multiple of p+1.   □

The above theorem shows the average time to rendezvous and the probability of rendezvous within a period of *p*-ary m-sequence. Based on the result of the above theorem, the ETTR for applying a *p*-ary m-sequence to an asymmetric channel can be determined as follows:

**Theorem 3.** 
*For positive integers M and p such that p is the smallest prime bigger than M. Let m(t) be a p-ary m-sequence with period $p^{2}-1$ and C be the number of common channels between both devices. Using m(t) as the channel-hopping sequence for the cognitive radio with M asymmetric channels, the ETTR of p-ary m-sequence for the cognitive radio with M asymmetric channels is $\left(p-2\right)+\frac{p(p-1-M)}{2M}$.*


**Proof.** From the result of Theorem 1, the average time to rendezvous of the *p*-ary m-sequence is p×(p−1−M)/(2M) with probability $\left(p^{2}-p+1\right)/\left(p^{2}-1\right)$. Since the two devices share *M* common channels out of a total of *p* channels, a rendezvous is guaranteed to occur after the (p−M+1)th search. Therefore, the probability of a successful rendezvous in the *i*-th period is $\left(\frac{p^{2}-p+1}{p^{2}-1}\right){\left(\frac{p-2}{p^{2}-1}\right)}^{i-1}$. According to the result of Theorem 2, if a rendezvous occurs in the *i*-th period, the ETTR for the *i*-th period is $\frac{p(p-1-M)}{2M}$. When rendezvous occurs in *i* th period, the ETTR is given as follows:
$$ETTR_{asym-ith}=\frac{p(p-1-M)}{2M}\left(\frac{p^{2}-p+1}{p^{2}-1}\right){\left(\frac{p-2}{p^{2}-1}\right)}^{i-1}+\left(i-1\right)\left(p^{2}-1\right){\left(\frac{p-2}{p^{2}-1}\right)}^{i-1}.$$Therefore, we can derive the ETTR as follows.
$$\begin{array}{ccc} ETTR_{asym} & = & \sum_{i=1}^{p-M+1}ETTR_{asym-ith}\\ & = & \sum_{i=1}^{p-M+1}\left\{\frac{p(p-1-M)}{2M}\left(\frac{p^{2}-p+1}{p^{2}-1}\right){\left(\frac{p-2}{p^{2}-1}\right)}^{i-1}+\left(i-1\right)\left(p^{2}-1\right){\left(\frac{p-2}{p^{2}-1}\right)}^{i-1}\right\}\\ & = & \sum_{i=0}^{p-M}\left\{\frac{p(p-1-M)}{2M}\left(\frac{p^{2}-p+1}{p^{2}-1}\right){\left(\frac{p-2}{p^{2}-1}\right)}^{i}+i\left(p^{2}-1\right){\left(\frac{p-2}{p^{2}-1}\right)}^{i}\right\}\\ . \end{array}$$The above expression can be calculated as follows:
$$\begin{array}{ccc} ETTR_{asym} & = & \frac{p(p-1-M)}{2M}\left(\frac{p^{2}-p+1}{p^{2}-1}\right)\frac{1-{\left(\frac{p-2}{p^{2}-1}\right)}^{p-M+1}}{1-(\frac{p-2}{p^{2}-1})}\\ & & +\left(p-2\right)\frac{1-\left(p-M+1\right){\left(\frac{p-2}{p^{2}-1}\right)}^{p-M+1}+\left(p-M+1\right){\left(\frac{p-2}{p^{2}-1}\right)}^{p-M+2}}{{\left(1-\left(\frac{p-2}{p^{2}-1}\right)\right)}^{2}}. \end{array}$$
If p−M is sufficiently large, ${\left(\frac{p-2}{p^{2}-1}\right)}^{p-M+1}$ and ${\left(\frac{p-2}{p^{2}-1}\right)}^{p-M+2}$ become negligible, allowing the approximations $1-{\left(\frac{p-2}{p^{2}-1}\right)}^{p-M+1}\approx 1$ and $1-\left(p-M+1\right){\left(\frac{p-2}{p^{2}-1}\right)}^{p-M+1}+\left(p-M+1\right){\left(\frac{p-2}{p^{2}-1}\right)}^{p-M+2}\approx 1$. Then, the above equation can be simplified as follows:
$$\begin{array}{ccc} ETTR_{asym} & \approx & \frac{p(p-1-M)}{2M}\left(\frac{p^{2}-p+1}{p^{2}-1}\right)\frac{1}{1-(\frac{p-2}{p^{2}-1})}+\left(p-2\right)\frac{1}{{\left(1-\left(\frac{p-2}{p^{2}-1}\right)\right)}^{2}}\\ & = & \frac{p(p-1-M)}{2M}\left(\frac{p^{2}-p+1}{p^{2}-1}\right)\frac{1}{\frac{p^{2}-p+1}{p^{2}-1}}+\left(p-2\right)\frac{1}{{\left(\frac{p-2}{p^{2}-1}\right)}^{2}-2\left(\frac{p-2}{p^{2}-1}\right)+1}\\ & = & \frac{p(p-1-M)}{2M}\left(\frac{p^{2}-p+1}{p^{2}-1}\right)\frac{1}{\frac{p^{2}-p+1}{p^{2}-1}}+\left(p-2\right)\frac{{\left(p^{2}-1\right)}^{2}}{{\left(p^{2}-1-p+2\right)}^{2}}\\ & = & \frac{p(p-1-M)}{2M}+\left(p-2\right){\left(\frac{p^{2}-1}{p^{2}-p+1}\right)}^{2}\\ & \approx & \frac{p(p-1-M)}{2M}+\left(p-2\right). \end{array}$$   □

The ETTR value derived from Theorem 3 does not significantly differ from the average TTR value for a rendezvous occurring within one cycle of the *p*-ary m-sequence in Theorem 2. Thus, from the result of Theorem 3, it can be concluded that the probability of rendezvous failure for the *p*-ary m-sequence, occurring when the time difference between the two devices is a multiple of p+1 has a small impact on the overall ETTR performance if the number of the common channel *M* is small.

## 3. Applying *p*-Ary m-Sequence to Reduced Asymmetric Channel Model

This section introduces a new channel-searching scheme optimized for *p*-ary m-sequences. By reducing the number of target channels, the proposed scheme aims to improve the ETTR while maintaining a high probability of rendezvous. We also derive the ETTR of the proposed scheme to validate its performance theoretically. Before discussing the new channel-searching scheme, we first need to look at the useful properties of the m-sequence.

**Definition 2** (Two-tuple balanced, Gong and Song [15]). *Let n be a positive integer and q be a power of prime number. Let s(t) be a q-ary sequence of period $q^{n}-1$, and let $T=\left(q^{n}-1\right)/\left(q-1\right)$. For a given integer τ with $0<\tau <q^{n}-1$ and $x,y\in F_{q}$, we define*
$$N\left(x,y\right)=\left|\{t|\left(s\left(t\right),s\left(t+\tau \right)\right)=\left(x,y\right),0\leq t<q^{n}-1\}\right|.$$*Then, s(t) is said to be two-tuple-balanced if we have $N\left(x,y\right)=q^{n-2}$ for (x,y)≠(0,0) with $N\left(0,0\right)=q^{n-2}-1$ when τ≢0(modT), and $N\left(x,y\right)=q^{n-1}$ for (x,y)≠(0,0) with $N\left(0,0\right)=q^{n-1}-1$ when τ≡0(modT).*

**Theorem 4** (Gong and Song [15]). *Let us assume that a q-ary sequence s(t) of period $q^{n}-1$ is difference-balanced and has the cyclic array structure. Then, it is two-tuple-balanced*

For a prime *p* and a positive integer *n*, it is well known that the *p*-ary m-sequence with period $p^{n}-1$ is difference-balanced. From Definition 1, Definition 2, and Theorem 4, we can say, the *p*-ary m-sequence with period $p^{n}-1$ is two-tuple balanced.

The conventional channel-searching scheme applied to the asymmetric channel model operates such that devices perform channel searching across all channels within the system, as shown in the following figure. When all channels achieve matching, a rendezvous occurs on a channel that is commonly available to both devices. See Figure 1.

As discussed in the introduction of this section, the m-sequence possesses a two-tuple balanced property. Therefore, if a *p*-ary m-sequence is used as the channel-searching pattern, each channel will attempt to match with all target channels of the other device, as in Figure 2.

In this paper, we propose a new channel-searching scheme that reduces the number of target channels and the frequency of searches for rendezvous by decreasing the number of target channels. To ensure the existence of a common channel, the number of target channels must exceed half of the total channels. Additionally, to apply the *p*-ary m-sequence, it is preferable for the number of target channels to be the smallest prime number greater than half of the total number of channels. The newly proposed channel-searching scheme in this paper is as follows:

**Theorem 5.** 
*For positive integer M, let N be an integer such that N=⌈M/2⌉ and q be a the smallest prime bigger than N. Let $m_{q}\left(t\right)$ be a q-ary m-sequence with period $q^{2}-1$ and the available channels of each device be assigned to the alphabets of the q-ary m-sequence in the order of the channel index, starting from 1. Furthermore, the best channel of each device is assigned to the 0 of the m-sequence. Using $m_{q}\left(t\right)$ as the channel-hopping sequence for the cognitive radio with M asymmetric channels, the rendezvous occurrs within $q^{2}-1$ with probability q/(q+1).*


**Proof.** From the properties of the *p*-ary m-sequence and Equation (Equation 2), it is clear that the *p*-ary m-sequence is difference-balanced and has the cyclic array structure. According to the results of Definition 2 and Theorem 3, it is evident that two *q*-ary m-sequences with a period of $q^{2}-1$ exhibit the two-tuple balance property when the phase difference τ between them is not a multiple of T=q+1. Therefore, when two devices utilize *q*-ary m-sequences as their channel-searching patterns, a rendezvous is guaranteed to occur within $q^{2}-1$ time slots, except in cases where the phase difference between the devices is a multiple of T=q+1. Since there are q−1 multiples of q+1 within a period of sequence, the property of rendezvous is q/(q+1).    □

From the result of the above theorem, we can derive the average time to rendezvous of the proposed scheme if the time delay between the two device is not a multiple of T=p+1 as follows.

**Theorem 6.** 
*For positive integer M, let N be an integer such that N=⌈M/2⌉ and q be a the smallest prime bigger than N. Let $m_{q}\left(t\right)$ be a q-ary m-sequence with period $q^{2}-1$ and C be the number of common channels between both devices. Using $m_{q}\left(t\right)$ as the channel-hopping sequence for the cognitive radio with M asymmetric channels, the ETTR of this scheme is $\left(q^{2}-1\right)/\left(2N\right)$ except where the time difference of the two devices is not a multiple of q+1.*


**Proof.** From the tuple-balanced-property in Theorem 4 and Equation (Equation 2), it is clear that the *q*-ary msequence has every pair $\left(a,b\right)\in \left(F_{q},F_{q}\right)\setminus \left\{\left(0,0\right)\right\}$. Therefore, the probability of every pair $\left(a,b\right)\in \left(F_{q},F_{q}\right)\setminus \left\{\left(0,0\right)\right\}$ is equal to $1/(q^{2}-1)$. From the assumption of this theorem, we have *N* common channels; rendezvous can occur *N* times within the period of the *q*-ary m-sequence. Therefore, the average TTR is $\left(q^{2}-1\right)/\left(2N\right)$ except when the time difference between the two devices is a multiple of q+1.    □

The above theorem shows the average time to rendezvous and the probability of rendezvous within a period of *q*-ary m-sequence. Based on the result of the above theorem, the ETTR for applying a *q*-ary m-sequence to an asymmetric channel with the proposed channel-searching scheme can be determined as follows:

**Theorem 7.** 
*For positive integer M, let N be an integer such that N=⌈M/2⌉ and q is the smallest prime bigger than N. Let $m_{q}\left(t\right)$ be a q-ary m-sequence with period $q^{2}-1$ and C be the number of common channels between both devices. Using $m_{q}\left(t\right)$ as the channel-hopping sequence for the cognitive radio with M asymmetric channels with the proposed scheme in Theorem 5, the ETTR of the q-ary m-sequence for the cognitive radio with M asymmetric channels is $q+\frac{q^{2}-1}{2N}$.*


**Proof.** From the result of Theorem 6, the average time to rendezvous of the *q*-ary m-sequence is $\left(q^{2}-1\right)/\left(2N\right)$ with probability q/(q+1). Since the two devices share *N* common channels out of a total of *q* channels, a rendezvous is guaranteed to occur by the (q−N+1)th search. Therefore, the probability of a successful rendezvous in the *i*-th period is $\left(\frac{q}{q+1}\right){\left(\frac{1}{q+1}\right)}^{i-1}$. According to the result of Theorem 6, if a rendezvous occurs in the *i*-th period, the ETTR for the *i*-th period is $\frac{q^{2}-1}{2N}$. When a rendezvous occurs in the *i* th period, the ETTR is given as follows:
$$ETTR_{pro-ith}=\frac{q^{2}-1}{2N}\left(\frac{q}{q+1}\right){\left(\frac{1}{q+1}\right)}^{i-1}+\left(i-1\right)\left(q^{2}-1\right)\left(\frac{q}{q+1}\right){\left(\frac{1}{q+1}\right)}^{i-1}.$$Therefore, we can derive the ETTR as follows:
$$\begin{array}{ccc} ETTR_{pro} & = & \sum_{i=1}^{q-N+1}ETTR_{pro-ith}\\ & = & \sum_{i=1}^{q-N+1}\left\{\frac{q^{2}-1}{2N}\left(\frac{q}{q+1}\right){\left(\frac{1}{q+1}\right)}^{i-1}+\left(i-1\right)\left(q^{2}-1\right)\left(\frac{q}{q+1}\right){\left(\frac{1}{q+1}\right)}^{i-1}\right\}\\ & = & \sum_{i=0}^{q-N}\left\{\frac{q^{2}-1}{2N}\left(\frac{q}{q+1}\right){\left(\frac{1}{q+1}\right)}^{i}+\left(i-1\right)\left(q^{2}-1\right)\left(\frac{q}{q+1}\right){\left(\frac{1}{q+1}\right)}^{i}\right\}. \end{array}$$The above expression can be calculated as follows:
$$\begin{array}{ccc} ETTR_{pro} & = & \frac{q^{2}-1}{2N}\left(\frac{q}{q+1}\right)\frac{1-{\left(\frac{1}{q+1}\right)}^{q-N+1}}{1-(\frac{1}{q+1})}\\ & & +\frac{q(q^{2}-1)}{{\left(q+1\right)}^{2}}\frac{1-\left(q-N+1\right){\left(\frac{1}{q+1}\right)}^{q-N+1}+\left(q-N+1\right){\left(\frac{1}{q+1}\right)}^{q-N+2}}{{\left(1-\left(\frac{1}{q+1}\right)\right)}^{2}}. \end{array}$$If q−M is sufficiently large, ${\left(\frac{1}{q+1}\right)}^{q-N+1}$ and ${\left(\frac{1}{q+1}\right)}^{q-N+2}$ become negligible, allowing the approximations $1-{\left(\frac{1}{q+1}\right)}^{q-N+1}\approx 1$ and $1-\left(q-N+1\right){\left(\frac{1}{q+1}\right)}^{q-N+1}+\left(q-N+1\right){\left(\frac{1}{q+1}\right)}^{q-N+2}\approx 1$. Then, the above equation can be simplified as follows:
$$\begin{array}{ccc} ETTR_{pro} & \approx & \frac{q^{2}-1}{2N}\left(\frac{q}{q+1}\right)\frac{1}{1-(\frac{1}{q+1})}+\frac{q(q^{2}-1)}{{\left(q+1\right)}^{2}}\frac{1}{{\left(1-\left(\frac{1}{q+1}\right)\right)}^{2}}\\ & = & \frac{q^{2}-q}{2N}\frac{1}{1-(\frac{1}{q+1})}+\frac{q(q^{2}-1)}{{\left(q+1\right)}^{2}}\frac{{\left(q+1\right)}^{2}}{q^{2}}\\ & = & \frac{q^{2}-q}{2N}\frac{q+1}{q}+\frac{\left(q^{2}-1\right)}{q}\\ & = & \frac{q^{2}-1}{2N}+\frac{\left(q^{2}-1\right)}{q}\\ & \approx & \frac{q^{2}-1}{2N}+q. \end{array}$$   □

The ETTR value derived from Theorem 7 does not significantly differ from the average TTR value for a rendezvous occurring within one cycle of the *q*-ary m-sequence in Theorem 1. Thus, from the result of Theorem 7, it can be concluded that the probability of rendezvous failure for the qp-ary m-sequence, occurring when the time difference between the two devices is a multiple of q+1 has a small impact on the overall ETTR performance if the number of common channels *N* is small

Let us consider an example for the case where *M* is relatively large.

**Example** **1.***Suppose that two devices in an asymmetric channel environment are attempting to achieve rendezvous. In this case, assume there are 100 channels in the system, with 30% (i.e., 30 channels) being shared between the two devices. Furthermore, assume that even when the newly proposed channel-searching scheme presented in this section is applied, the number of shared channels remains 30% of the total channels used. When using the m-sequence as the channel-searching pattern, the ETTR values for the existing method and the newly proposed method can be calculated as follows, based on the results of Section 3 and Section 4:* 
p=101,q=53,M=30,N=15.
$$\begin{array}{ccc} ETTR_{asym} & = & \frac{p(p-1-M)}{2M}+\left(p-2\right)=\frac{101(100-30)}{2\times 30}+99=216.83\\ ETTR_{pro} & = & \frac{q^{2}-1}{2N}+q=\frac{53^{2}-1}{2\times 15}+53=146.6 \end{array}$$

Let us consider another example for the case where *M* is 1.

**Example** **2.***Suppose that two devices in an asymmetric channel environment are attempting to achieve rendezvous. In this case, assume there are 100 channels in the system, with 1 (i.e., 30 channels) being shared between the two devices. Furthermore, assume that even when the newly proposed channel-searching scheme presented in this section is applied, the number of shared channels remains 1 of the total channels used. When using the m-sequence as the channel-searching pattern, the ETTR values for the existing method and the newly proposed method can be calculated as follows, based on the results of Section 3 and Section 4.* 
p=101,q=53,M=1,N=1.
$$\begin{array}{ccc} ETTR_{asym} & = & \frac{p(p-1-M)}{2M}+\left(p-2\right)=\frac{101(100-1)}{2\times 1}+99=5098.5\\ ETTR_{pro} & = & \frac{q^{2}-1}{2N}+q=\frac{53^{2}-1}{2\times 1}+53=1457 \end{array}$$

As seen in the two examples above, the newly proposed channel-searching scheme demonstrates significantly better performance compared to the conventional scheme when the number of common channels is small.

## 4. Numerical Results

In this section, we compare the rendezvous performance of the *p*-ary m-sequence for the asymmetric model with the conventional scheme and the newly proposed scheme. To evaluate the performance across various environments, we will conduct simulations, increasing the number of total channels from 30 to 100, each tested with 30% common channels and 1 common channel, respectively. For performance comparison, the simulations will compare three schemes: the JS algorithm [6], the conventional scheme using a *p*-ary m-sequence, and the newly proposed scheme using a *q*-ary m-sequence. The JS algorithm for the asymmetric channel model is defined as follows:

**Definition 3** (JS Algorithm for Asymmetric Channel Model [6]). *Let M, the total number of channels, be a positive integer. Then, the JS algorithm for the asymmetric channel is given as Algorithm 1.*
**Algorithm 1** JS Algorithm for Asymmetric Channel Model1: **Input**: *M*2: *p* = *the smallest prime number greater than M*;3: *r*_0_ = *rand*[1,*M*]; *i*_0_ = *rand*[1,*M*]; *t* = 0;4: **WHILE** (*not rendezvous*)5:     *n* = ⌊*t*/3*p*⌋; *r* = (*r*_0_ + *n* − 1) *mod*
*M* + 1;6:     *n* = ⌊*t*/6*Mp*⌋; *i* = (*i*_0_ + *n* − 1) *mod p* + 1;7:     *c* = *JSHopping*(*M*, *p*, *r*, *i*, *t*); *t* = *t* + 1;8:     *Attempt rendezvous on channelc*;9: **END**


Figure 3 illustrates the simulation results when 30% of the total channels are shared. The simulations, based on the average TTR over 10,000 rendezvous events, reveal that the proposed scheme outperforms the conventional scheme and the JS algorithm. Specifically, the ETTR of the proposed scheme is approximately half that of the conventional scheme.

Figure 4 shows the results when only one common channel exists. In this extreme scenario, the proposed scheme demonstrated ETTR values roughly 25% of those achieved by the conventional scheme. This improvement is attributed to the reduced cycle length of the *q*-ary m-sequence used in the proposed scheme.

## 5. Conclusions

In this paper, we propose a channel-searching pattern and a searching scheme for achieving rendezvous in an asymmetric channel model. The newly proposed channel-searching scheme leverages the properties of the m-sequence to maximize rendezvous performance, as demonstrated through both theoretical analysis and simulation results. Additionally, we derive and compare the theoretical ETTR values for applying the m-sequence to a conventional channel-searching scheme and to the newly proposed scheme in the asymmetric channel model. The simulation results show that applying the *p*-ary m-sequence to the conventional channel-searching scheme significantly outperforms the JS algorithm, both when the number of common channels is 30% and when there is only one common channel. This superior performance is attributed to the fact that, despite the m-sequence having a certain failure probability, the average time to rendezvous is very low within a single cycle. Moreover, applying the m-sequence to the newly proposed channel-searching scheme yields even better performance compared to the conventional scheme. Notably, when there is only one common channel, the proposed scheme achieves ETTR values that are nearly 25% that of the conventional scheme. This improvement is attributed to the m-sequence’s characteristic of achieving high rendezvous probability within a single cycle, even when the number of channels to be searched is reduced.

## 6. Patents

The results of this will be submitted as patent within this year.

## Figures and Tables

**Figure 1 sensors-25-00062-f001:**
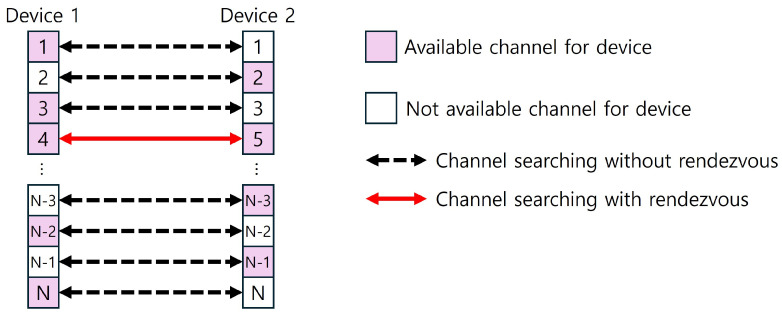
Conventional channel-searching scheme for asymmetric channel model.

**Figure 2 sensors-25-00062-f002:**
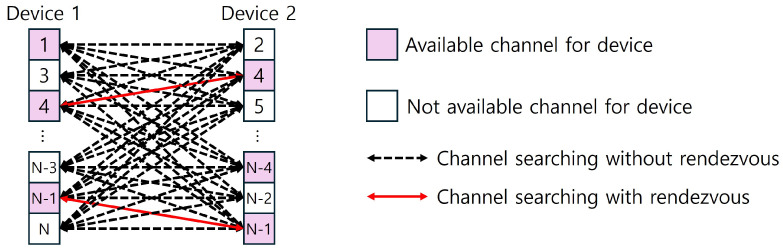
Proposed channel-searching scheme for asymmetric channel model.

**Figure 3 sensors-25-00062-f003:**
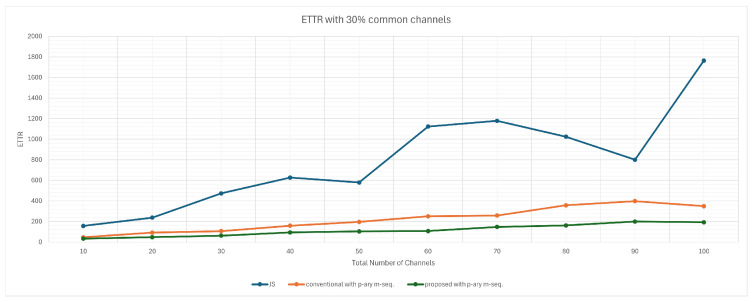
ETTR with 30% common channels.

**Figure 4 sensors-25-00062-f004:**
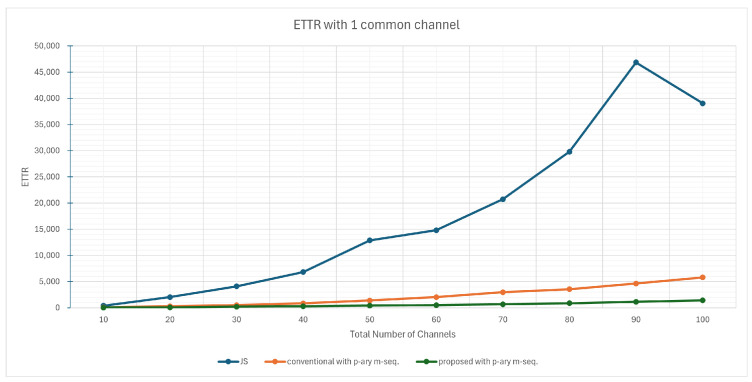
ETTR with 1 common channel.

## Data Availability

No new data were created or analyzed in this study. Data sharing is not applicable to this article.

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
