# Peer review of "Channel-Hopping Sequence and Searching Algorithm for Rendezvous of Spectrum Sensing"

_sensors, 2024, doi:10.3390/s25010062_

Round 1
Reviewer 1 Report
Comments and Suggestions for Authors
This paper provides a new solution for the channel hopping and rendezvous problems in cognitive radio networks, and verifies its effectiveness through theoretical analysis and simulation. Especially in the asymmetric channel model, it discusses how to apply the p-ary m-sequence as a channel hopping sequence to achieve fast rendezvous. The paper mathematically analyzes the expected time to rendezvous (ETTR) when the m-sequence is applied to the conventional scheme, and demonstrates through simulation results that its performance is significantly better than the JS algorithm. Finally, it also proposes possible directions for future research, including further optimizing channel search algorithms and exploring the potential of m-sequence in other applications. It provides a good direction for future research in related fields. This paper closely follows current research hotspots, with innovative ideas, clear logic, and standardized writing. It can be accepted after minor revision as follows.
1. The text in Figures 3 and 4 is not clear enough and needs to be modified.
2. It may be necessary to add units to the axes of Figures 3 and 4.
3. The article may need to describe the JS algorithm in more detail so that readers can make a better comparison.
4. There appear to be encoding issues with the numbers (e.g.124, 150, 169) that are not displayed correctly. These should be addressed to avoid any confusion.
5. The article should either include the experimental results of the JS algorithm in Figure 4 or provide a more detailed explanation for its exclusion.
Author Response
Comment 1. The text in Figures 3 and 4 is not clear enough and needs to be modified.
- The text for Figure 3 and Figure 4 has been revised.
Comment 2. It may be necessary to add units to the axes of Figures 3 and 4.
- Units have been added to the axes of Figure 3 and Figure 4.
Comment 3. The article may need to describe the JS algorithm in more detail so that readers can make a better comparison.
- Details on the generation method of the JS algorithm in the asymmetric channel model have been added to the Numerical Results section.
Comment 4. There appear to be encoding issues with the numbers (e.g.124, 150, 169) that are not displayed correctly. These should be addressed to avoid any confusion.
- I apologize, but I am unable to understand the question clearly. The numbers 124, 150, and 169 do not appear in my manuscript.
Comment 5. The article should either include the experimental results of the JS algorithm in Figure 4 or provide a more detailed explanation for its exclusion.
- The performance of the JS algorithm has been added to Figure 4.
Reviewer 2 Report
Comments and Suggestions for Authors
This paper, titled "Channel Hopping Sequence and Searching Algorithm for Rendezvous of Spectrum Sensing," introduces a novel approach utilizing the p-ary m-sequence method to enhance the efficiency of rendezvous in asymmetric channel models of cognitive radio networks. The authors provide theoretical analyses and simulations to support the proposed scheme, demonstrating improvements in Expected Time to Rendezvous (ETTR) over conventional and JS algorithm-based methods. While the study offers valuable insights, I have identified several questions and suggestions that could help the authors improve the clarity, depth, and overall quality of the manuscript. These points are aimed at refining the presentation and addressing potential ambiguities in the methodology and results.
1. Some mathematical proofs (e.g., Theorem 3) are dense and challenging to follow. Could the authors provide step-by-step breakdowns or append additional explanations in the supplementary material?
2. The paper provides theoretical results, but how does the proposed method perform under real-world conditions, including noise, interference, or hardware limitations? Are there any assumptions that may not hold in practical deployments?
3. The simulations use specific parameters (e.g., 30% shared channels or 1 shared channel). Could the authors elaborate on why these scenarios were chosen and how representative they are of real-world applications?
4. The reduced channel scheme depends on the smallest prime greater than half the total channels. How does this choice impact systems with a large number of channels? Are there scalability concerns?
5. The analysis mentions failure probabilities when the phase difference between devices is a multiple of \(p+1\). How often does this occur in practice, and what mitigation strategies could be applied?
6. The JS algorithm is referenced but not included in one of the graphs (Figure 4) due to variability. Could the authors include an average performance metric for JS for consistency in comparisons?
7. How does your proposed method compare to the approach presented in the paper *"A New Strategy for Dynamic Channel Allocation in CR-WMN Based on RCA"* in terms of key performance metrics such as channel utilization, time to rendezvous, scalability, and adaptability to dynamic network conditions? Additionally, could you provide insights into the specific advantages or limitations of your method relative to the RCA-based strategy mentioned in that paper?
8. The conclusion highlights future research directions. Could the authors specify the next steps for testing and refining their method, such as hardware testing or integration with existing communication standards?
9. Figures 3 and 4 illustrate results, but could additional visual aids or flow diagrams be included to clarify the working of the proposed schemes for readers unfamiliar with p-ary sequences?
These questions aim to enhance the presented research's clarity, applicability, and broader understanding.
Comments on the Quality of English Language
Here are some grammatical and spelling mistakes found in the article:
1. "This case is occurs only once for \(p^2 - 1\) cases." can be corrected as: "This case occurs only once for \(p^2 - 1\) cases."
2. "In this case, randezvous is not occured." can be corrected as: "In this case, rendezvous does not occur."
3. "We should another trial to randezvous for this case." can be corrected as: "We should make another trial to rendezvous for this case."
4. "Let devide \(p^2 - 1\) time slots to \(p - 1\) sets..." can be corrected as: "Let divide \(p^2 - 1\) time slots into \(p - 1\) sets..."
5. "The average TTR of (q² − 1)/(2N) except for the time difference of two devices is not the multiple of \(q + 1\)." can be corrected as: "The average TTR is \((q^2 - 1)/(2N)\), except when the time difference between two devices is a multiple of \(q + 1\)."
6. "Therefore, rendezvous is occured with a probability of \((p^2 − p + 1)/(p^2 − 1)\)." can be corrected as: "Therefore, rendezvous occurs with a probability of \((p^2 - p + 1)/(p^2 - 1)\)."
7. "Rendezvous can be occured \(N\) times within the period..." can be corrected as: "Rendezvous can occur \(N\) times within the period..."
8. "We can describe the the value of trace function..." can be corrected as: "We can describe the value of the trace function..."
These corrections improve the grammatical accuracy of the article.
Author Response
Comment 1. Some mathematical proofs (e.g., Theorem 3) are dense and challenging to follow. Could the authors provide step-by-step breakdowns or append additional explanations in the supplementary material?
- I have added some more steps on the proof of Theorem 3 and Theorem 7.
Comment 2. The paper provides theoretical results, but how does the proposed method perform under real-world conditions, including noise, interference, or hardware limitations? Are there any assumptions that may not hold in practical deployments?
- The focus of this paper is on the performance of channel hopping patterns. Rendezvous serves as the initial connection between devices in cognitive radio (CR) systems that aim to communicate. Subsequent optimal channel selection for communication is achieved using information exchanged through the channel established during rendezvous. Therefore, the quality of the channel connected via rendezvous has minimal impact on actual communication. Consequently, the primary objective of a channel hopping pattern is to locate and establish a common channel as quickly as possible, which is central to its performance evaluation.
Comment 3. The simulations use specific parameters (e.g., 30% shared channels or 1 shared channel). Could the authors elaborate on why these scenarios were chosen and how representative they are of real-world applications?
- The 30% shared channels case was used because it is a commonly employed scenario in channel hopping pattern simulations. The case with only one shared channel was included to demonstrate the performance in an extreme scenario.
Comment 4. The reduced channel scheme depends on the smallest prime greater than half the total channels. How does this choice impact systems with a large number of channels? Are there scalability concerns?
- There are two reasons for using the smallest prime greater than half the total channels in the reduced channel scheme. First, the $p$-ary m-sequence is generated based on prime numbers. Second, using a value greater than half the total channels ensures that shared channels always exist. In cognitive radio systems, the assumption of shared channels is typically made; however, unlike conventional methods that consider the entire set of total channels, the newly proposed scheme operates on a reduced set of channels. Therefore, it is essential to guarantee the presence of shared channels. By using the smallest prime greater than half the total channels, the proposed scheme ensures that there are always shared channels between the two devices.
Comment 5. The analysis mentions failure probabilities when the phase difference between devices is a multiple of \(p+1\). How often does this occur in practice, and what mitigation strategies could be applied?
- As stated in the paper, the period of the $p$-ary m-sequence used as the channel hopping pattern is $p^2 -1$. As you can see in the result of Theorem 1, the failure probability is given by $(p-2)/(p^2-1)$. As shown in Theorem 1 and Theorem 5, the proposed method in the paper has a non-zero failure probability. However, compared to conventional methods, the proposed method has a shorter hopping pattern period and a smaller ETTR upon success. To address the possibility of failing to rendezvous after two full cycles of the sequence, the method employs a phase shift to retry. The ETTR reflecting this approach is provided in Theorem 3 and Theorem 7 of the paper. In other words, Theorem 3 and Theorem 7 account for ETTR under the scenario where failures occur due to specific phase differences.
Comment 6. The JS algorithm is referenced but not included in one of the graphs (Figure 4) due to variability. Could the authors include an average performance metric for JS for consistency in comparisons?
- The performance of the JS algorithm has been added to Figure 4.
Comment 7. How does your proposed method compare to the approach presented in the paper *"A New Strategy for Dynamic Channel Allocation in CR-WMN Based on RCA"* in terms of key performance metrics such as channel utilization, time to rendezvous, scalability, and adaptability to dynamic network conditions? Additionally, could you provide insights into the specific advantages or limitations of your method relative to the RCA-based strategy mentioned in that paper?
- The paper"A New Strategy for Dynamic Channel Allocation in CR-WMN Based on RCA" focuses on minimizing interference and improving network throughput in Cognitive Radio Networks (CRNs). Specifically, it addresses the challenges of interference in Multi-Radio Multi-Channel (MRMC) networks by analyzing and mitigating both coordinated interference and uncoordinated interference. The primary goal is to propose a dynamic channel allocation strategy that leverages unsupervised machine learning to efficiently manage frequency resources and optimize network transmission performance.
In contrast, the this paper aims to design an efficient channel searching pattern to achieve rendezvous in an asymmetric channel model. It leverages the desirable properties of the p-ary m-sequence, such as its ideal autocorrelation and uniform distribution, to propose a new algorithm that significantly reduces the Expected Time to Rendezvous (ETTR). The focus is on ensuring stable and rapid rendezvous even in asymmetric environments with varying channel availability.
Comment 8. The conclusion highlights future research directions. Could the authors specify the next steps for testing and refining their method, such as hardware testing or integration with existing communication standards?
- At present, there are no plans for hardware testing or testing under existing communication standards.
Comment 9. Figures 3 and 4 illustrate results, but could additional visual aids or flow diagrams be included to clarify the working of the proposed schemes for readers unfamiliar with p-ary sequences?
- Providing a detailed explanation of the generation of the $p$-ary m-sequence would require the inclusion of extensive material on finite fields, shift registers, and primitive polynomials, which would significantly lengthen the manuscript. We apologize, but we believe it is more appropriate to refer readers to Reference 7, 'Encyclopedia of Mathematics and Its Applications', for detailed information on m-sequence generation.
These questions aim to enhance the presented research's clarity, applicability, and broader understanding.
Comments on the Quality of English Language
Here are some grammatical and spelling mistakes found in the article:
Comment 1. "This case is occurs only once for \(p^2 - 1\) cases." can be corrected as: "This case occurs only once for \(p^2 - 1\) cases."
- I corrected it according to the reviewer's comment
Comment 2. "In this case, randezvous is not occured." can be corrected as: "In this case, rendezvous does not occur."
- I corrected it according to the reviewer's comment
Comment 3. "We should another trial to randezvous for this case." can be corrected as: "We should make another trial to rendezvous for this case."
- I corrected it according to the reviewer's comment
Comment 4. "Let devide \(p^2 - 1\) time slots to \(p - 1\) sets..." can be corrected as: "Let divide \(p^2 - 1\) time slots into \(p - 1\) sets..."
- I corrected it according to the reviewer's comment
Comment 5. "The average TTR of (q² − 1)/(2N) except for the time difference of two devices is not the multiple of \(q + 1\)." can be corrected as: "The average TTR is \((q^2 - 1)/(2N)\), except when the time difference between two devices is a multiple of \(q + 1\)."
- I corrected it according to the reviewer's comment
Comment 6. "Therefore, rendezvous is occured with a probability of \((p^2 − p + 1)/(p^2 − 1)\)." can be corrected as: "Therefore, rendezvous occurs with a probability of \((p^2 - p + 1)/(p^2 - 1)\)."
Comment 7. "Rendezvous can be occured \(N\) times within the period..." can be corrected as: "Rendezvous can occur \(N\) times within the period..."
- I corrected it according to the reviewer's comment
Comment 8. "We can describe the the value of trace function..." can be corrected as: "We can describe the value of the trace function..."
- I corrected it according to the reviewer's comment